# Coupling of 3-Aminopropyl Sulfonic Acid to Cellulose Nanofibers for Efficient Removal of Cationic Dyes

**DOI:** 10.3390/ma15196964

**Published:** 2022-10-07

**Authors:** Naglaa Salem El-Sayed, Ahmed Salama, Vincenzo Guarino

**Affiliations:** 1Cellulose and Paper Department, National Research Centre, 33 El-Bohouth St., Dokki, Giza P.O. Box 12622, Egypt; 2Institute of Polymers, Composites and Biomaterials, National Research Council of Italy, Mostra d’Oltremare, pad.20, V.le Kennedy 54, 80125 Naples, Italy

**Keywords:** cellulose nanofibers, adsorption, cationic dyes

## Abstract

A novel anionic nanostructured cellulose derivate was prepared through the coupling of TEMPO-oxidized cellulose nanofibers with 3-aminopropyl sulfonic acid (3-APSA). 3-APSA grafting was variously investigated by FT-IR spectroscopy and transmission electron microscopy (TEM) analysis, confirming a high reaction degree. The surface morphology investigated via scanning electron microscopy (SEM) revealed a more uniform organization of the nanofibers after the 3-APSA coupling, with improvements in terms of fiber packing and pore interconnectivity. This peculiar morphology contributes to improving methylene blue (MB) adsorption and removal efficiency at different operating conditions (pH, initial time, and initial concentration). The results indicated a maximum adsorption capacity of 526 mg/g in the case of 3-APSA grafted nanofibers, over 30% more than that of non-grafted ones (370 mg/g), which confirm a relevant effect of chemical modification on the adsorbent properties of cellulose nanofibers. The adsorption kinetics and isotherms of the current adsorbents match with the pseudo-second-order kinetic and Langmuir isotherm models. This study suggests the use of chemical grafting via 3-APSA is a reliable and facile post-treatment to design bio-sustainable and reusable nanofibers to be used as high-performance adsorbent materials in water pollutant remediation.

## 1. Introduction

Cellulose is the most abundant natural polymer on earth, providing a sustainable green resource that is renewable, degradable, and biocompatible, with cost-effectiveness [1,2,3]. Cellulose and its derivatives have been studied for their application as adsorbents and membranes for the purification and detoxification of water due to their excellent adsorptive capacity [4,5]. The limited solubility of cellulose in water and organic solvents affects its chemical reactivity and limits its use in different applications. The solubility of cellulose can be improved by its chemical modification, e.g., tosylation [6,7,8], TEMPO oxidation [9,10], esterification, or etherification [11]. These reactions have been widely applied to broaden their utilization in many environmental, technological, and biomedical applications, although different approaches have been suggested for their chemical modifications. In particular, the selective oxidation of cellulose using 2,2,6,6-tetramethylpiperidine-1-oxyl (TEMPO) is one of the most influential and demanding approaches for the selective conversion of the primary alcohol groups at C6 into the corresponding aldehydes or carboxylic acid groups [12]. In addition, it offers a high reaction rate with minimal chain degradation throughout the process. TEMPO-oxidized cellulose nanofibers (TOCNF) provide a thrilling route for introducing new carboxyl groups that can effectively facilitate the immobilization of new small molecules, proteins, peptides, or macromolecules [13,14,15].

On the other hand, heavy metals, phenolic compounds, dyestuffs, and colorant agents are the most common organic water pollutants. With the escalation of consumer demands and industrial development, the risk of water pollution with such contaminants increases, which in turn negatively influences the ecosystem and threatens public health [16,17,18,19]. 

There are different bio-adsorbents that have been proposed for the removal of organic dyes, such as anionic methyl orange (MO) and cationic methylene blue from aqueous solutions. Among these bio-absorbents are activated carbons (ACs), which are derived from the waste of consumed food or drinks [20]. For instance, Block et al. [21] reported the production of ACs through the pyrolysis of sodium-hydroxide-pretreated dried spent coffee waste (SCD), which exhibited nonselective adsorption for methyl orange (MO) and cationic methylene blue. Meanwhile, Nguyen et al. [22] reported the synthesis of ACs from teak sawdust. The resulted ACs efficiently adsorbed methylene blue in addition to Cd(II) and Cu(II) ions. Also, agriculture waste such as sugar cane bagasse and rice strew were used as sustainable resources for the preparation of carbon-based materials, such as biochar and cellulose nanofibers (CNFs). CNFs have been identified as highly functional and bio-based adsorbents for different organic and inorganic pollutants due to their small fiber diameters (>10 nm), high aspect ratios (>100), high specific surface areas, abundant functionalities, high crystallinities (65–95%), high tensile strengths (200–300 MPa), high elastic moduli (6–7 GPa), and low ζ-potentials in water (ca. −75 mV). Moreover, different methodologies were proposed to functionalize NCFs to improve their adsorbent capacity and selectivity [4,23]. For instance, when polyethyleneimine (PEI) was grafted to TOCNF, the resulting hydrogel dramatically increased the initial adsorption capacity of Cu (II) adsorption, i.e., achieved a- maximum Cu(II) uptake of 52.32 mg/g—as confirmed by comparative kinetic studies with samples without PEI [24]. Zhu et al. optimized a membrane with high hydrolytic stability and recyclability by blending TOCNF with graphene oxide (GO), acting as an adsorbent for Cu(II) ions. The results showed that GO did not only boost the removal capacity of Cu(II) ions, but also increased the membrane flexibility, hydrolytic stability, and mechanical strength [25]. Similarly, Liu et al. designed cellulose beads by mixing TOCNF with Fe(III) ions via the extrusion dropping technique. In this case, (Fe-CCB) beads were used to spontaneously remove the bromide ions from water at low concentrations under neutral or acidic conditions due to their nonfibrillar structure [26]. In another study, titanate@TOCNF-reinforced chitosan hydrogels, modified with fluorescent carbon dots, showed an efficient adsorption ability for Cr (VI) ions. In this case, fluorescent hydrogel showed a maximum adsorption capacity for Cr(VI) at 228.2 mg/g with a detection limit at a concentration of 10–80 mg/L [27]. TOCNF was used for the adsorption of cationic metal ions, but it was also used to address the detoxification of organic pollutants and water remediation. In this regard, Hussain et al. fabricated a series of recyclable hybrid monolith GO/TOCNF composites for the adsorption of methylene blue (MB) dye. The adsorbent completely removed the traces of MB dye following pseudo-second-order kinetics, attaining a maximum adsorption capacity of 227.27 mg/g [28]. Accordingly, the in situ preparation of TiO_2_ nanoparticles using TOCNF resulted in the formation of TOCNF/TiO_2_ nanocomposites, which displayed a maximum adsorption capacity for brilliant blue dye (BB) of up to 162 mg/g at pH 7.

On the other hand, 1,1′-carbonyldiimidazole (CDI) has been used as a green, solvent-free activator to accelerate other coupling reactions such as esterification, amidation, and carbamate reactions [29]. In particular, CDI offers several advantages over other organic or phosphor-organic activators due to its strong electrophilic nature. Indeed, it can activate carboxylic acid, alcohol, or amine groups [30,31]. CDI is extremely cheap compared with other activating reagents and can be easily produced in kilos. Further, its use promotes the delivery of harmless by-products such as CO_2_ gas and imidazoles, which act as base catalysts during coupling reactions. Therefore, CDI is commonly used in pharmaceutical and fine chemical industries for the scalable manufacturing of a variety of aliphatic, aromatic, and heterocyclic products [32].

In this work, we developed a novel chemical method for the efficient and direct preparation of new bio-adsorbent materials to remove methylene blue dye. The preparation involved using TEMPO-oxidized CNFs produced from sugar cane bagasse pulp. Then, CDI activation for the generated carboxylic groups along the TOCNF chains accelerated the rate of the amidation reaction with 3-APSA. Chemical and physical characterizations were performed to validate the sustainable and green properties of 3-APSA-grafted TOCNF for the facile fabrication of scalable bio-adsorbents.

## 2. Materials and Methods

### 2.1. Materials

3-Aminopropyl sulfonic acid (APSA, with 98% purity) was purchased from VWR International and used as-received. Bleached bagasse pulp was supplied from Qena Company (city of Qena, Egypt) of Paper Industry, Egypt. The chemical composition of the bleached bagasse pulp was 69.23% alpha-cellulose, 29% pentosans, 0.92% lignin, and 0.85% ash content. 2,2,6,6-tetramethylpiperidine-1-oxyl (TEMPO, 95%) and 1,1′-carbonyldiimidazole (CDI, 97%) were purchased from Merck.

### 2.2. Methods

#### 2.2.1. Preparation of TEMPO-Oxidized Cellulose Nanofiber (TOCNF1)

TEMPO-oxidized cellulose nanofiber was prepared from bagasse pulp according to the method described by Salama et al. [12]. In brief, 20 g of bleached bagasse pulp was dispersed in distilled water with TEMPO (0.8 g) and sodium bromide (8 g). Then, 300 mL of sodium hypochlorite solution (15%) was added with continuous stirring, and the pH was adjusted to 10.0 using the NaOH (3 mol/L) solution. The pH was adjusted to 7.0, and the product was centrifuged at 10,000 rpm several times. Finally, the product was purified with dialysis for 1 week against deionized water.

#### 2.2.2. Coupling of 3-Aminopropyl Sulfonic Acid APSA to TOCNF1

TOCNF1 (30 g) was mixed with 1 g of DIC and ball-milled in a mortar for 10 min. Then, APSA (0.3 g, 7.1 mmole), dissolved in 50 mL of water, was added dropwise to the DIC-activated TOCNFs. The reaction was continually stirred at 75 °C for 72 h. Afterward, the fibers were separated by filtration and washed three times with boiling water (3 × 50 mL) and three times with boiling ethanol (3 × 50 mL) to ensure the complete removal of unreacted APSA and DCI. Finally, the fibers that had the code name TOCNF2 were collected and freeze-dried for 48 h.

### 2.3. Characterization

#### 2.3.1. Elemental Analysis

The elemental analyses for TOCNF1 and TOCNF2 were recorded with Vario Elementar (Germany) in triplicate.

#### 2.3.2. Fourier Transfer Infrared Spectroscopy

The FT-IR spectra of TOCNF1 and TOCNF2 were recorded with a FT-IR spectrometer (Nicolet Impact-400 FT IR spectrophotometer, Thermo Scientific, Madison, WI) (in the range of 400–4000 cm^−1^.

#### 2.3.3. Scanning Electron Microscopy (SEM)

The surface morphology of TOCNF1 and TOCNF2 were analyzed using scanning electron microscopy (JSM 6360LV, JEOL/Noran). The microscope was attached to an energy dispersive X-ray analysis (EDX) unit. The samples for SEM were freeze-dried for 72 h before imaging. The images were obtained using an accelerating voltage of 10–15 kV.

#### 2.3.4. Transmittance Electron Microscopy (TEM)

Transmission electron microscopy (TEM) for TOCNF1 and TOCNF2 was recorded with a JEOL JEM-2100 electron microscope at 100k× magnification, with an acceleration voltage of 120 kV.

#### 2.3.5. Adsorption Experiments

The study of the influence of the contact time (10–120 min), initial concentration (50–800 mg/L), and pH (3–8) on the adsorption of methylene blue (MB) was performed in batch experiments. In brief, 25 mg of TOCNF1 and TOCNF2 was added at room temperature into MB solutions, and the pH of the latter was adjusted using HCl (0.1 M) and NaOH (0.1 M) solutions. After adsorption, the supernatants were separated from the mixture through centrifugation at a constant speed (3000 rpm) for 5 min. The aliquots were withdrawn from the suspension and measured at maximum absorbance (670 nm) using a spectrophotometer (UNICO UV-2000). Maximum adsorption capacity (*qe*) was obtained by the following equation:(1)$$qe=\frac{\left(Ci-Co\right)V}{m}$$
where *Ci* (mg/L) and *Co* (mg/L) represent the original and equilibrium dye concentrations, respectively. The parameter *qe* demonstrates maximum adsorption capacity at equilibrium. The volume of dye solution is represented by *V* (L), and the mass of the adsorbent film serves as m (g). The desorption of MB from CNF2 was evaluated utilizing an HCl solution. The dried sample was dispersed in the eluent using a thermostatic bath shaker operated at 160 rpm for 24 h. The cyclic adsorption−desorption was completed five times to determine the reusability of the adsorbent. The standard deviation was added. Data are representative of at least three experiments, and the standard deviations were less than 7.0%.

## 3. Results and Discussion

The TEMPO-mediated oxidation of cellulose is an efficient reaction method for the selective oxidation of the primary hydroxyl groups at C-6 of cellulose into carboxylic groups. The generated carboxyl groups at C-6 are very reactive and can undergo further functionalization reactions [9]. TOCNF1 coupled to 3-aminopropane sulfonic acid APSA through the formation of stable amide linkages between the free amino groups of APSA and the carboxylic groups at C6 of TOCNF1. In this case, 1,1′-carbonyldiimidazole (CDI) was used as an activating agent. The carboxyl groups of TONCF1 were activated with CDI to form a reactive intermediate of TOCNF-acyl imidazole that was able to react with the free amine groups (nucleophiles) from APSA via pure amide bonds, while the imidazole served as a base for the coupling reaction (Figure 1).

Figure 2 shows the FT-IR spectra of TOCNF1 and modified TOCNF2. The two spectra showed characteristic bands around 3450 cm^−1^ that could be attributed to OH stretching vibrations and the inter- and intra-molecular hydrogen bindings. This peak was significantly higher and sharper in the case of TOCNF2 due to the presence of additional amide groups. Additionally, the increased intensity of the adsorption band at 1590 cm^−1^ in the case of TOCNF2 suggested the creation of an amide group moiety. The -CH asymmetric and symmetric stretching vibration peaks of the methylene groups were seen around 2917 and 2858 cm^−1^. As a result of 3-APSA coupling to TOCNF1, the distinctive peaks for the symmetric and asymmetric S-O stretching vibration of the sulphonic groups could be observed at 1342 (υas SO2), 1159 (υs SO2), and 839 cm^−1^ (νS–O–C) [6].

Figure 3 shows the surface morphology of TOCNF1 and modified TOCNF2 investigated with SEM. The morphological analysis showed a nanofibrillar structure of TOCNF1 with a random spatial organization and an inconsistent distribution of fiber diameters (Figure 3A). After grafting with APSA, a more homogeneous spatial organization of fibers was recognized. Moreover, the modification of CNFs through amide bond formation with APSA increased the fiber packing and the network interconnectivity due to the increased inter-chain hydrogen bonds (Figure 3B). Additionally, the elemental composition estimated with EDX (point and small area measurements) showed an EDX spectra for TONCF2 including the characteristic peaks for nitrogen and sulfur atoms with atomic percents of 4.2 and 1.3, respectively (Figure 3C).

TEM analysis was also performed to further investigate morphological features at the nanoscale. In the case of TOCNF1, fibers with length and the average diameters varying from 4 to 20 nm were calculated via image analysis (Figure 3D). There were no relevant differences in size, but a more uniform nanorod-like structure was recognized in the case of TOCNF2 (Figure 3E).

In order to evaluate the capacity of anionic TOCNF1 and TOCNF2 as adsorbents for cationic dyes, MB was chosen and investigated in detail as a model for cationic dyes and the adsorption process.

First, the cationic dye adsorption capacity of TOCNF1 and TOCNF2 at different pH levels was assessed. Figure 4 shows that an extensive increase in the removal efficiency was achieved in the case TOCNF2 compared with that of TONCF1. Moreover, the attitude to MB removal as the pH values increased was higher for TOCNF2 compared with that for TOCNF1, as was also confirmed by the maximum adsorption amount acquired for pH values over six. The adsorption of MB at pH 6 gradually increased up to a record of 85 and 91 mg/g for TOCNF1 and TOCNF2, respectively. Overall, TOCNF2 exhibited a higher adsorption capacity than TOCNF1. This performance proved the role of the sulphonyl groups in TOCNF2, which offered additional sulfonic functional groups that contributed to the other functional groups in coordination with the cationic MB. The adsorption capacity for both materials reached a plateau at pH values between six and eight. This behavior can be explained as follows: when the pH level was below six, higher concentrations of H+ ions occurred and directly competed with the dye molecules for chelation with anionic functional groups [29]. Additionally, a further increase of the solution acidity determined the sulphonic group shift to –SO3H rather than –SO3ˉ. By all the experimental analyses, the pH study revealed a maximum adsorption capacity at pH ˃ 7. Consequently, pH 7 was chosen as the optimum pH value to conduct the additional adsorption experiments [33].

As for the measurement of the adsorption capacity of TOCNF1 and TOCNF2, an evaluation of the adsorption equilibrium was required. Figure 5 clearly shows that the adsorption capacity of MB onto TOCNF1 and TOCNF2 fibers at different times rapidly increased at the initial stage. TOCNF reached an adsorption efficiency of 88 mg/g, which corresponded to an MB equilibrium time of 60 min. However, when TOCNF1 was functionalized with 3-APSA, its efficiency increased to 94 mg/g. This result confirmed that the presence of sulfonic groups improved the adsorption capacity of the modified cellulose nanofibers. Other studies were performed by adjusting the saturation adsorption at 60 min fixed as the equilibrium time.

Moreover, during the ion adsorption, solutes were transferred from the liquid phase to the surface of the solid phase. For a better understanding of the kinetics, pseudo-first-order (Equation (2)), pseudo-second-order (Equation (3)), and Elovich kinetic models (Equation (4)) were adopted.
(2)$$\log\;\left(q_{e\;}-q_{t\;}\right)=\log(q_{e\;})-\frac{K_{1}}{2.303}t$$
(3)$$\frac{t}{q_{t}}\;\;\frac{t}{q_{e}}\;\frac{1}{K_{2}q_{e}^{2}}$$
(4)$$q_{t}=\frac{\;1}{\beta }\;\mathrm{lin}\;(\alpha \beta )+\frac{1}{\beta }\ln t$$

Here, *q_t_* and *q_e_* are the amount of the adsorbed dye at the given time *t* (min) and equilibrium in milligrams per gram, respectively; *K*_1_ and *K*_2_ are the equilibrium rate constants of these two models, respectively. Further, α and β are the initial adsorption rate and desorption constant, respectively.

The calculated kinetic parameters by linear fitting for the adsorption of MB on the prepared TOCNF2 were analyzed with the most common kinetic models, i.e., the pseudo-first-order and -second-order ones. The adsorption data were modeled using first- and second-order kinetic equations. Table 1 displays the experimental and fitting data for TOCNF1 and TOCNF2. The kinetic adsorption of MB for the two samples was better fitted by the pseudo-second-order model, yielding R = 0.97 for the two samples. The pseudo-second-order constant, labeled *K*_2_, was 5.1 × 10^−4^ and 0.1 × 10^−2^ g mg^−1^ min^−1^ for TOCNF1 and TOCNF2, respectively. The calculations revealed that MB was adsorbed via chemisorption onto the prepared composite. Moreover, the Elovich kinetic model, which assumes the solid adsorbent surface to be energetically heterogeneous, didn’t match the experimental results very well [34].

In this context, the initial dye concentration showed an essential role during the adsorption mechanism. Figure 6 proves the effect of MB concentration on the adsorption capacity. In the case of MB content equal to 800 ppm, the uptake capacity increased to 340 and 481 mg/g for TOCNF1 and TOCNF2, respectively. As the MB concentration increased, an increase of the driving force to transfer MB from the aqueous to solid phase was attended due to the improved contact among MB molecules and cellulose nanofibers.

The correlation of equilibrium results by means of a theoretical or empirical equation is important for the adsorption clarification and expectation of the extent of adsorption [35]. Langmuir isotherms assume a homogeneous adsorption surface, the sites are identical, the adsorption occurs through monolayers, and no interaction occurs between adjacent adsorbate molecules [36,37]. The isotherm equation can be described as follows:(5)$$\frac{C_{e}}{q_{e}}=\frac{K_{s}}{q_{\max}}+\frac{C_{e}}{q_{\max}}$$
where *q_e_* is the equilibrium adsorption capacity (milligrams per gram), *q*_max_ is the maximum adsorption capacity (milligrams per gram), *C_e_* is the equilibrium concentrations of MB (milligrams per liter), and *K_s_* is the Langmuir model constant (milligrams per liter). Figure 6 describes the adsorption isotherms of MB for TOCNF1 and TOCNF2. The results showed that TOCNF2 had a good adsorption performance to MB. Langmuir and Freundlich adsorption isotherm models were used to fit the experimental data. Table 2 displays the adsorption isotherm parameters. The results showed that the adsorption of MB by TOCNF1 and TOCNF2 was more consistent with the Langmuir model (based on the R^2^ value), which showed that the active sites on the cellulose nanofibers were homogeneous, and the adsorption of MB was mostly a monolayer. Moreover, the values of the calculated constants, kilolitres and newtons, were between zero and one, indicating that the adsorption process was very favorable. The cellulose derivatives, CNF1 and CNF2, had negative charges in the aqueous solution due to the presence of carboxylate and sulphate groups. This could significantly enhance the electrostatic interaction between the adsorbents and adsorbates. These results indicated that the “as-developed cellulose nanofibers through the modification of 3-aminopropyl sulfonic acid coupling” approach was suitable for pollutant removal. From the slope and the intercept of the straight line, the values of *q*_max_ were estimated to be 526 and 370 mg/g, and the values of *K_s_* were 19.7 and 15.2 mg/L for TOCNF1 and TOCNF2, respectively.

The Freundlich and Temkin models are presented in Equations (6) and (7), respectively [38].
(6)$$\log q_{e}=\frac{1}{n}\log C_{e}+\log p$$
(7)$$q_{e}=\frac{RT}{b_{t}}\;\ln\;K_{T}+\frac{RT}{b_{t}}\;\ln\;C_{e}$$
where 1/*n* is the empirical parameter of the Freundlich model and *p* is a constant describing the adsorption capacity (milligrams per gram) and *B_T_* is the Temkin constants and *B_T_* = *RT*/*b_t_* (Joules per mole).

In applying the Freundlich model, which is based on adsorption on a heterogeneous surface, we found that the linear coefficient was 0.87 and 0.86, which indicated that this model did not describe the adsorption processes of TOCNF1 and TOCNF2 for MB very well (Table 2). The values of the Freundlich model constant were *p* (44 and 61), and n was 2.5 and 2.4 for TOCNF1 and TOCNF2, respectively. As shown in Table 2, the Temkin model did not match the experimental data very closely. This model (Temkin) describes the heterogeneous surface of functionalized cellulose with adsorption sites and assumes that adsorption sites on an adsorbent surface have different activities and can be described by a continuous distribution of activities.

The feasibility of the adsorption process was calculated using the separation factor (RL), which is defined by the following equation [39]:(8)$$R_{L}=\frac{1}{1+KsC_{O}}$$
where *C_o_* is the initial concentration of MB (parts per million). Based on this factor, an adsorption system can be predicted to be favorable or unfavorable. *R_L_* values for the adsorption of MB at different initial concentrations fell in the range of zero to one, according to the calculated results.

One of the most important properties of practical adsorbents is their reusability. The adsorption/desorption experiment was repeated five times, especially for TOCNF2, and the removal efficiency of MB was recorded to be 91% in the first cycle and 74% in the fifth cycle, as shown in Figure 7. Thus, the results indicated the reusability of the functionalized TEMPO-oxidized cellulose nanofibers.

From the results, it is possible to conclude that TOCNF2 is a promising material for MB absorption due to its higher adsorption capacities in comparison with those of other adsorbents based on polysaccharide-modified cellulose nanofibers (Table 3).

## 4. Conclusions

This paper describes the promising use of the pulp from sugar cane bagasse as a renewable sustainable resource and its conversion into carboxylated cellulose nanofibers via TEMPO-mediated oxidation. The resulting TOCNF1 reacted with 3-APSA following activation by CDI. The kinetic studies revealed that TOCNF2 was more efficient than TOCNF1 in terms of MB removal, showing a maximum adsorption capacity 1.4 times higher than that of TOCNF1 (without 3-APSA grafting). In this view, the use of chemical grafting via 3-APSA can be considered a reliable strategy to design bio-sustainable nanofibers to be used as high-performance adsorbent materials. From this perspective, the unique chance to process these materials by simple routes to design high performance devices with various forms (i.e., membranes, sheets, papers, foams, aerogels, etc.) paves the way for innovative and sustainable uses for water remediation.

## Figures and Tables

**Figure 1 materials-15-06964-f001:**
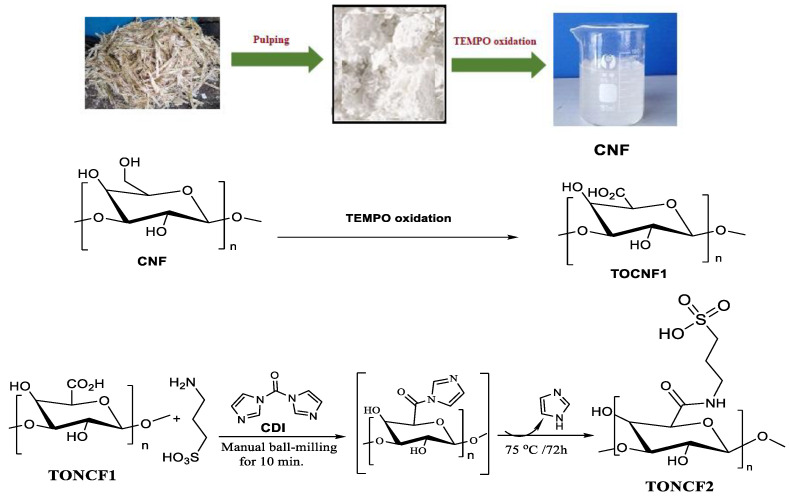
Extraction of cellulose pulp from sugar cane bagasse and its conversion into CNF. Oxidation of CNF by TEMPO affording TOCNF1. Then, the coupling between 3-APSA and TOCNF1 to generate TOCNF2.

**Figure 2 materials-15-06964-f002:**
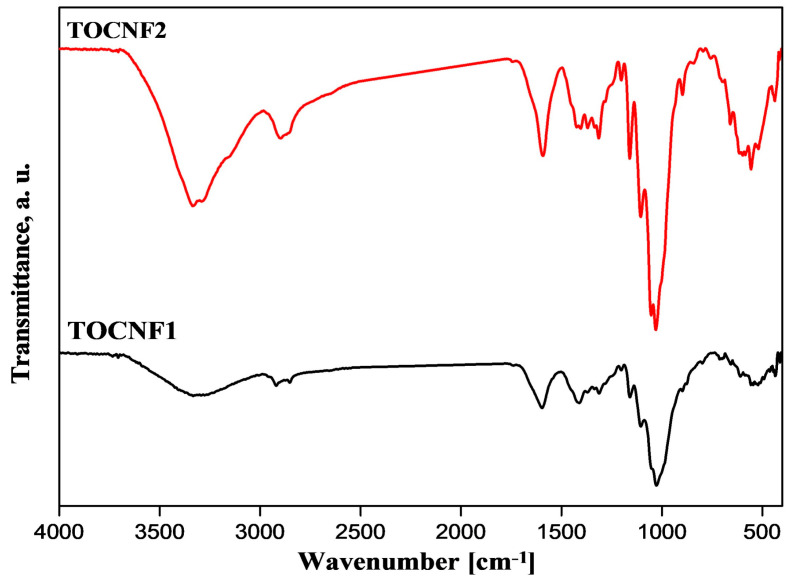
FT-R spectroscopy of TOCNF1 and modified TOCNF2.

**Figure 3 materials-15-06964-f003:**
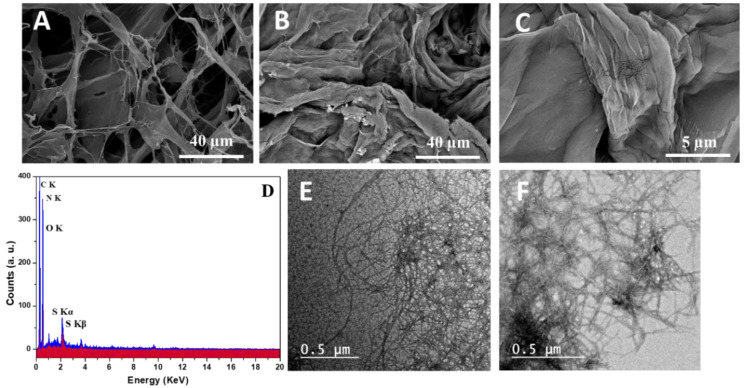
SEM images (**A**–**C**); EDX spectra (**D**) and TEM images (**E**,**F**) for TOCNF1 (**A**,**E**) and modified TOCNF2 (**B**–**F**).

**Figure 4 materials-15-06964-f004:**
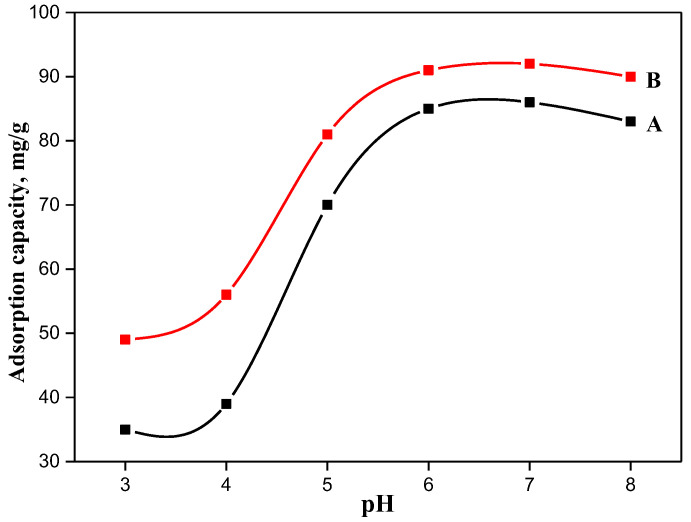
Effect of pH on the adsorption capacities of TOCNF1 (A) and TOCNF2 (B) towards MB (MB concentration: 100 mg/L; functionalized cellulose: 0.05 g/50 mL and contact time: 40 min).

**Figure 5 materials-15-06964-f005:**
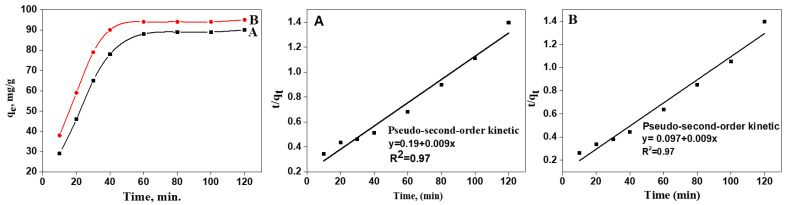
Effect of the adsorption time on adsorption capacity, and the adsorption kinetic fittings of TOCNF1 (**A**) and TOCNF2 (**B**).

**Figure 6 materials-15-06964-f006:**
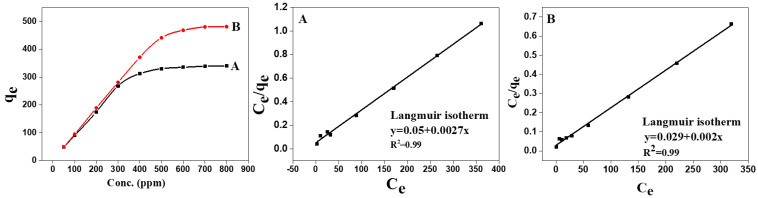
Effect of the initial MB concentration on adsorption capacity and Langmuir adsorption isotherm models for TOCNF1 (**A**) and TOCNF2 (**B**).

**Figure 7 materials-15-06964-f007:**
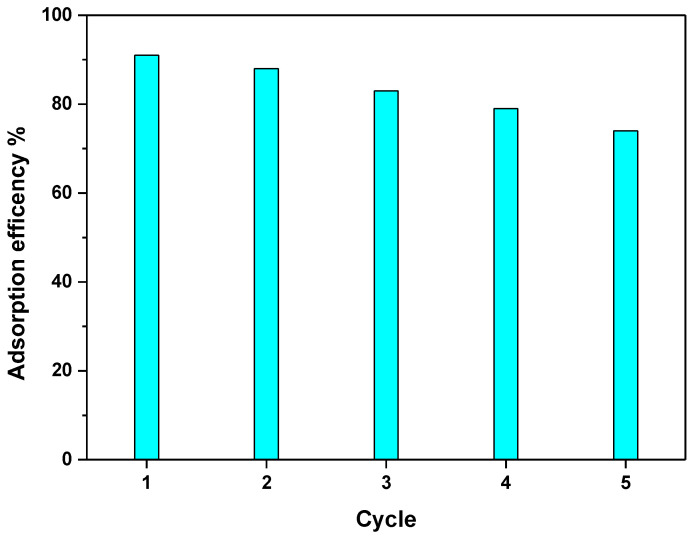
Adsorption efficiency of MB in five adsorption–desorption cycles for CNF2.

**Table 1 materials-15-06964-t001:** Kinetic parameters for MB adsorption for TOCNF1 and TOCNF2.

Models	Parameters	TCNF1	TCNF2
Pseudo-first-order	*q_e, cal_* (mg/g)	59	60
	*K*_1_ (min^−1^)	0.022	0.04
	R^2^	0.81	0.76
Pseudo-second-order	*q_e, cal_* (mg/g)	101	101
	*K*_2_ (g mg^−1^min^−1^)	5.1 × 10^−4^	10.3 × 10^−4^
	R^2^	0.97	0.97
Elovich	α	9.2	17.44
	β	0.04	0.04
	R^2^	0.91	0.84

**Table 2 materials-15-06964-t002:** Parameters for MB adsorption by TOCNF1 and TOCNF2 according to different equilibrium models.

Models	Parameters	TCNF1	TCNF2
Langmuir	*q_m_* (mg g^−1^)	370	526
	*K_s_* (L mg^−1^)	19.07	15.2
	R^2^	0.99	0.99
Freundlich	*P* (mg/g)	44	61
	*n*	2.5	2.4
	R^2^	0.87	0.86
Temkin	*B_T_* (J mol^−1^)	63.13	88.21
	*K_T_* (L mg^−1^)	0.91	1.2
	R^2^	0.9	0.91

**Table 3 materials-15-06964-t003:** Maximum adsorption capacity of methylene blue by modified polysaccharides.

Adsorbent	Maximum Adsorption (mg/g)	References
Cellulose grafted with soy protein isolate/calcium phosphate	454	[40]
Alginate/gelatin hydrogel-decorated silver nanoparticles	625	[41]
Cellulose/silk fibroin/calcium phosphate biocomposite	172.4	[42]
Chitosan/silica nanocomposite	848	[43]
Graphene/TEMPO-oxidized cellulose nanofiber	227.3	[28]
TOCNF2	526	Here

## Data Availability

Not applicable.

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
