# Peer review of "Coupling of 3-Aminopropyl Sulfonic Acid to Cellulose Nanofibers for Efficient Removal of Cationic Dyes"

_materials, 2022, doi:10.3390/ma15196964_

Round 1
Reviewer 1 Report
The manuscript concerns a novel cellulose-based adsorbent for methylene blue removal from water. This work is scientifically sound and the main required experiments and analyses were performed. Data are presented clearly and the paper is well-structured. In sum, I think the manuscript can be considered for publication in its present form.
Author Response
Thank you very much for your positive comments
Reviewer 2 Report
In this manuscript (materials-1927243), the authors prepared 3-APSA grafted cellulose nanofibers and investigated the adsorption ability of Methylene blue. The authors did much work. However, some issues need to be major revised before considering acceptance.
(1) The zeta potential of TOCNF1 and TOCNF2 should add, which help explore the adsorption mechanism.
(2) The BET analysis should add.
(3) Elovich adsorption kinetics models are recommended to add. Please refer to the paper (10.1016/j.cej.2022.138127).
(4) In adsorption isotherm, Temkin models should be added. Please refer to the paper (10.1016/j.jhazmat.2021.127192).
Author Response
In this manuscript (materials-1927243), the authors prepared 3-APSA grafted cellulose nanofibers and investigated the adsorption ability of Methylene blue. The authors did much work. However, some issues need to be major revised before considering acceptance.
(1) The zeta potential of TOCNF1 and TOCNF2 should add, which help explore the adsorption mechanism.
The zeta potential for the tow samples recorded very close results (-44 mV)
(2) The BET analysis should add.
The BET for oxidized cellulose nanofibers were carried out and the results showed that the value of the surface area is 64.7 m2/g. Moreover, this value doesn’t change significantly after chemical modification of cellulose.
(3) Elovich adsorption kinetics models are recommended to add. Please refer to the paper (10.1016/j.cej.2022.138127).
Elovich adsorption kinetics model was introduced in the article according to this reference (10.1016/j.cej.2022.138127).
(4) In adsorption isotherm, Temkin models should be added. Please refer to the paper (10.1016/j.jhazmat.2021.127192).
Temkin model was designed according to this article (10.1016/j.jhazmat.2021.127192) and the fitting constants were now introduced.
Reviewer 3 Report
In the present study, cellulose nanofibers were modified using 3-aminopropyl sulfonic acid and used as an absorbent to remove cationic dyes. The present study has favorable results and the presented material is well organized. It is suggested to be accepted after the following corrections:
1. In the abstract section, information about the optimal conditions of the surface adsorption process should also be presented. Also, materials about kinetic and isotherm results should be presented.
2. In the introduction section, there is no information about colors and their effects on the environment and their removal methods. It is suggested that in the introduction section, some information about colored pollutants should also be provided.
3. In section 2.1, it is necessary to present the materials used and their purity percentage completely.
4. In line 138, it is mentioned that the concentration of colors has been checked in the range of 50-800 mg/L. Is there a wastewater in industries that discharges this amount of paint into the environment? The reason for checking this concentration range should be justified.
5. In Figure 2, it is necessary to present the functional groups related to each peak.
6. pHzpc should be determined and the effect of pH should be investigated using it.
7. In the investigation of kinetic behavior and isotherm, it is necessary and necessary to use Dubinin-Radeshkovich and Temkin isotherm models and kinetic models of inter-particle diffusion and Elovich. It was suggested that the section on kinetic and isotherm studies be improved using the following articles.
• doi.org/10.1016/j.ijbiomac.2021.08.144
• doi.org/10.1016/j.jece.2021.106344
• doi.org/10.1016/j.chemosphere.2021.131088
8. The necessary mechanism for the process of surface absorption of cationic dyes using the desired adsorbent should be presented.
9. The ability to regenerate and reuse the desired adsorbent in removing cationic dyes should be checked.
Author Response
Reviewer 3
In the present study, cellulose nanofibers were modified using 3-aminopropyl sulfonic acid and used as an absorbent to remove cationic dyes. The present study has favorable results and the presented material is well organized. It is suggested to be accepted after the following corrections:
1. In the abstract section, information about the optimal conditions of the surface adsorption process should also be presented. Also, materials about kinetic and isotherm results should be presented.
Information about optimal condition, kinetic and isotherm results were included in the abstract.
2. In the introduction section, there is no information about colors and their effects on the environment and their removal methods. It is suggested that in the introduction section, some information about colored pollutants should also be provided.
We thank the reviewer for the comment aimed to improve the overall quality of the manuscript. The authors have modified the introduction part, adding some recent papers on the hazardous effect of the dyes and colorant on the environment.
3. In section 2.1, it is necessary to present the materials used and their purity percentage completely.
The authors are grateful for the reviewer constructive comments, the authors have included the % purity for the chemicals used in section 2.1.
4. In line 138, it is mentioned that the concentration of colors has been checked in the range of 50-800 mg/L. Is there a wastewater in industries that discharges this amount of paint into the environment? The reason for checking this concentration range should be justified.
Thank you for this comment. The text has been amended accordingly.
5. In Figure 2, it is necessary to present the functional groups related to each peak.
Thank you for the comment, the description of results reported in Figure 2 has been improved in order to better understanding the reported data.
6. pHzpc should be determined and the effect of pH should be investigated using it.
Thank you for this comment. This evaluation will be performed in future studies on these materials, referring to specific environmental conditions as a function of the application of interest.
7. In the investigation of kinetic behavior and isotherm, it is necessary and necessary to use Dubinin-Radeshkovich and Temkin isotherm models and kinetic models of inter-particle diffusion and Elovich. It was suggested that the section on kinetic and isotherm studies be improved using the following articles.
- doi.org/10.1016/j.ijbiomac.2021.08.144
- doi.org/10.1016/j.jece.2021.106344
- doi.org/10.1016/j.chemosphere.2021.131088
The kinetic behaviour and the isotherms were investigated with additional models according to the reviewers and the references were cited.
8. The necessary mechanism for the process of surface absorption of cationic dyes using the desired adsorbent should be presented.
Thank you for your comment. From Langmuir firing the adsorption process is chemosorption and additional discussion about the adsorption mechanism was included in the text article. Cellulose derivatives, CNF1 and CNF2, have negative charges in the aqueous solution due to the presence of carboxylate and sulphate groups. This can significantly enhance the electrostatic interaction between the adsorbent and adsorbate.
9. The ability to regenerate and reuse the desired adsorbent in removing cationic dyes should be checked.
The ability of reusability was discussed and included in the article. The modified cellulose showed high degree of reusability
Reviewer 4 Report
Dear Authors
This manuscript is focused on the novel anionic nanostructured cellulose derivate, which was prepared through the coupling of TEMPO oxidized cellulose nanofibers with 3-aminopropyl sulfonic acid (3-APSA).
The following suggestion and comments should be taken:
1. The overall English needs to be improved. Please seek guidance from a native English speaker if possible ("the" "a", commas, plural form and others could be corrected).
2. The introduction section needs enhancement few sentences about others dyes used with carbon materials. Please cite (1) Materials 2021, 14(14), 3996; https://doi.org/10.3390/ma14143996 (2) Materials 12 (20), 3354, 2019 https://doi.org/10.3390/ma12203354 (3) Water 2019, 11(12), 2581; https://doi.org/10.3390/w11122581
3. Could the authors include the standard deviation of used methods?
4. Figure 2. Please correct this image for better quality.
5. Figure 3 SEM. Please add more images with different magnification.
6. Figure 5. Please correct this image for better quality.
7. Figure 6. Please correct this image for better quality.
8. Please add some sentences about difference Langmuir-Freudlich isotherms.
9. Authors are suggested to describe some future plans in conclusions.
Author Response
Dear Authors
This manuscript is focused on the novel anionic nanostructured cellulose derivate, which was prepared through the coupling of TEMPO oxidized cellulose nanofibers with 3-aminopropyl sulfonic acid (3-APSA).
The following suggestion and comments should be taken:
1: The overall English needs to be improved. Please seek guidance from a native English speaker if possible ("the" "a", commas, plural form and others could be corrected).
The authors are grateful for the reviewer constructive comments, the authors have checked the manuscript and corrected all the typo and grammatic mistakes.
2. The introduction section needs enhancement few sentences about others dyes used with carbon materials. Please cite (1) Materials 2021, 14(14), 3996; https://doi.org/10.3390/ma14143996 (2) Materials 12 (20), 3354, 2019 https://doi.org/10.3390/ma12203354 (3) Water 2019, 11(12), 2581; https://doi.org/10.3390/w11122581
The authors are grateful for the reviewer suggestion; we have enriched the introduction section with the valuable references suggested by the reviewer.
3. Could the authors include the standard deviation of used methods?
The standard deviation was added. Data are representative of at least three experiments, and standard deviations are less than 8.0%.
4. Figure 2. Please correct this image for better quality.
The image was improved to better quality
5. Figure 3 SEM. Please add more images with different magnification.
The resolution of the image was improved, and additional images were added
6. Figure 5. Please correct this image for better quality.
The quality of figure 5 was improved
7. Figure 6. Please correct this image for better quality.
The quality of figure 6 was improved
8. Please add some sentences about difference Langmuir-Freudlich isotherms.
The discussion of kinetic and isotherm study was improved
9. Authors are suggested to describe some future plans in conclusions.
Thank you for your question. A sentence has been included to add a perspective of future activities.
Round 2
Reviewer 2 Report
accept!
Reviewer 4 Report
Now work is ok